# Study Designs for Evaluation of Combination Treatment: Focus on Individual Patient Benefit

**DOI:** 10.3390/biomedicines10020270

**Published:** 2022-01-26

**Authors:** Martin C. Michel, David Staskin

**Affiliations:** 1Department of Pharmacology, Johannes Gutenberg University, 55131 Mainz, Germany; 2St. Elizabeth Medical Center, School of Medicine, Boston University, Boston, MA 02135, USA; staskinatt@g-mail.com

**Keywords:** combination treatment, monotherapy, clinical trial design, benefit/risk assessment

## Abstract

Combination treatment, i.e., the use of two or more drugs for the same condition, is frequent in medicine if monotherapy yields an insufficient therapeutic response. We review and challenge clinical study designs and formats of reporting outcomes for the evaluation of the benefit/risk ratio of combination treatment over monotherapy. We demonstrate that benefits of combination treatment at the group level overestimate the probability of benefit at the single patient level based on outcome simulations under almost any imaginable setting. Based on these findings, we propose that studies testing combination treatment should always report on percentages of responders to monotherapy and combination treatment. We provide equations that allow the calculation of the percentage of patients truly benefitting from combination (responders to both monotherapies) and that of patients exposed to risk of harm from adverse effects without a reasonable expectation of individual benefit. These considerations are explained based on real clinical data, mostly from the field of functional urology (male lower urinary tract symptoms).

## 1. Introduction

Combination treatment, i.e., the concomitant use of two or more drugs for the same indication, is frequently considered when administering a single drug yields an insufficient therapeutic response. Guideline-recommended examples of these include the combination of a diuretic and a Ca^2+^ entry blocker in the treatment of arterial hypertension [1]; of an inhibitor of the renin-angiotensin system, a β-adrenoceptor antagonist, a mineralocorticoid receptor antagonist and a sodium-glucose transporter 2 inhibitor in the treatment of heart failure with reduced ejection fraction [2]; of a glucocorticoid and a β_2_-adrenoceptor agonist in the treatment of asthma [3]; of various types of agents in oncology [4]; of a muscarinic receptor antagonist and a β_3_-adrenoceptor agonist in the treatment of overactive bladder syndrome [5]; or of an α_1_-adrenoceptor antagonist and a 5α-reductase inhibitor in the treatment of male lower urinary tract symptoms (LUTS) attributed to benign prostatic hyperplasia [6]. This paper aims to review and challenge the rationale behind frequently used study designs and reporting formats in the evaluation of combination treatment as compared to monotherapy. We will illustrate the discussed concepts largely with examples from the field of functional urology (male LUTS) but propose that our considerations will be applicable to many if not most areas of medicine; a systematic review of possible examples was neither performed nor intended. We propose that our considerations apply regardless of whether the efficacy of combination partners is additive, less than additive, or even more than additive.

## 2. General Considerations, Current Practice, and Regulatory Recommendations

### 2.1. General Considerations

A meaningful combination treatment should fulfill several general criteria. Firstly, the co-administered drugs typically should not affect the pharmacokinetic profile of each other to avoid pharmacokinetic drug–drug interactions. Exceptions are combinations explicitly designed in a manner, such as the combination of a β-lactam antibiotic and a lactamase inhibitor, to yield greater exposure to the β-lactam, for instance amoxicillin + clavulanic acid. Second, the drugs being combined should not exhibit additive toxicology as also highlighted by regulatory guidelines [7]. Third, the combination partners should exhibit compatible pharmacokinetic profiles, particularly if administration in a single pill is intended as a fixed-dose combination (FDC).

### 2.2. Paralell Group vs. Add-On Study Designs

Most evaluations of potential drug combinations are based on one of two types of study designs. One type uses parallel groups of one or more monotherapies and combination treatment and has, for instance, been applied to compare the effects of an α_1_-adrenoceptor antagonist, a 5α-reductase inhibitor, and their combination in the treatment of male LUTS [8,9,10,11,12]. An advantage of this approach is that it is not biased by considerations on which drug to administer first and which to add. However, a disadvantage is that starting all regimens in parallel does not allow determining which individuals (patients) need combination therapy, and it also includes subjects who already show a sufficient response to one of the monotherapies. Thus, this approach is suitable to explore whether the addition of a novel drug provides benefit on top of the standard of care. It is also suitable when effects on long-term, particularly major outcomes such as mortality, are investigated, where waiting for non-responder status would be too late, unethical, and/or unpractical for timeline reasons. For instance, a parallel group design has been used to determine whether sodium-glucose transporter 2 inhibitors dapagliflozin [13] and empagliflozin [14] reduce mortality and other cardiovascular endpoints in patients with heart failure with reduced rejection fraction when added on top of the standard of care.

A second approach is based on selecting insufficient responders to a monotherapy and then giving a second drug as add-on treatment. The main benefit of this approach is that it focuses on patients with an insufficient response to monotherapy, i.e., those who may need combination treatment. In its simplest manner, this can be performed by open-label addition of a second drug. This has been applied, for instance, in a study exploring the effects of adding the β_3_-adrenoceptor agonist mirabegron to the treatment of male LUTS with the 5α-reductase inhibitor dutasteride [15]. Its key limitation is that the interpretation of outcomes is limited by various biases including a possible placebo effect and/or observer bias. Thus, in this example, studies comparing mirabegron alone with placebo have demonstrated that approximately 60% of the response were also observed in the placebo group [16]. Of note, the placebo effect itself exhibits a ‘dose–response’ curve: In placebo-controlled dose-escalation studies, giving a second placebo to those exhibiting limited treatment effects to a single dose increased the perceived effect [17]. Due to its limitations, this design can be informative in the context of a pilot study but should not be seen as more than generating a signal in which a combination may be worth another look. Another possible application of this approach is open-label, post-marketing authorization studies in which non-responders to an established drug are given a recently approved drug, for instance, mirabegron on top of the muscarinic receptor antagonist solifenacin [18]. This approach can be useful if the benefit of the combination has already been established and investigators wish to determine how it manifests in real-life practice.

To establish the efficacy of a combination as compared to monotherapy, it is more informative to randomize insufficient responders to either continue with monotherapy or to receive the second drug, as reported for instance in a study where consecutive men with LUTS receiving the α_1_-adrenoceptor antagonist tamsulosin and not exhibiting a sufficient improvement were randomized to additionally receive the muscarinic receptor antagonist tolterodine or not [19]. While this approach reduces biases to some extent by having a parallel control group, it remains open to vulnerability in terms of placebo effect and observer bias and, if anything, is also largely limited to pilot studies. Therefore, the gold standard of the add-on design is to randomize insufficient responders to receive placebo or the second drug in a double-blind manner. This yields trials of similar high quality as other placebo-controlled, double-blind, randomized studies. Therefore, this is frequently applied in studies for regulatory purposes. Several variations of this approach exist, and all of them have been used in the clinical development program for a combination of mirabegron and solifenacin. For instance, it is possible to compare the effects of an add-on drug to both continuation (to control for time effects) and to a dosage increase in the first drug [20]. Alternatively, multiple doses of the second drug can be added in non-responders to an established treatment; this creates the add-on form of a dose-selection study. This may be more effective in identifying the right dose of combination partners than a parallel group design; for instance, one study has used 12 parallel arms including six with various doses of the combination to establish optimal doses of combinations of mirabegron and solifenacin [21].

Thus, each of these designs for the evaluation of combination treatments has distinct advantages and limitations. Parallel group studies are easy to design but imply that some adequate responders to one drug are unnecessarily exposed to a second drug. However, they are important in settings where long-term outcomes such as progression of male LUTS [11,12] or effects in conditions with a high mortality such as heart failure are studied [13,14]. On the other hand, various types of add-on designs can be appropriate to specifically target non-responders to one drug. It is telling that a single clinical development program for a combination of mirabegron and solifenacin has concomitantly used many of the above approaches. This may reflect an uncertainty in terms of which study design is the most appropriate and/or the realization that each of them may have specific advantages and disadvantages. Therefore, we propose that studies testing possible combination therapies should carefully weigh the advantages and disadvantages of parallel group vs. add-on designs to select the most appropriate format for the testing of a given combination.

### 2.3. Additional Considerations for Study Design

Determining an appropriate study design for combination treatment is further complicated by the fact that the benefit of a combination may not be equally applicable at all time points, particularly if time-to-onset for the desired therapeutic effects can be expected to differ between the combination partners. For instance, combination of the 5α-reductase inhibitor finasteride with the phosphodiesterase 5 inhibitor tadalafil as compared to finasteride + placebo yielded a group difference of 1.7, 1.4, and 1.0 International Prostate Symptom Score (IPSS) points after 4, 12, and 26 weeks, respectively [22]. This decline in difference was not unexpected because improvements by 5α-reductase inhibitors are known to require 3–6 months to become noticeable, whereas those of phosphodiesterase 5 inhibitors typically manifest within days. On the other hand, studies in men with LUTS and a study duration of up to 1 year reported that combination treatment with an α_1_-adrenoceptor antagonist (alfuzosin, doxazosin and terazosin) and a 5α-reductase inhibitor (finasteride) was not more efficacious than monotherapy with the α_1_-adrenoceptor antagonist [8,9,10], whereas studies with a duration of at least 2 years found combination treatment (doxazosin/finasteride or tamsulosin/dutasteride) to be more efficacious than either monotherapy at time points later than 1 year after the initiation of treatment [11,12].

Another consideration is that the statistical superiority of combination treatment as compared to monotherapy does not always translate into clinically relevant improvement. For instance, it is proposed that the improvement of the AUA Symptom Score/IPSS must be at least 3 points in order to be noticeable by a man with LUTS [23]. Other definitions of a responder in this indication includes a reduction in IPSS by ≥50% [9] or by ≥25% and ≥3 points [12]. However, studies with the combination of tamsulosin and solifenacin reported that the combination treatment using 6 or 9 mg solifenacin was statistically superior to monotherapy with tamsulosin based on about 300 patients per study arm, but the effect size was moderate at best (IPSS reduction by tamsulosin, tamsulosin +6 mg solifenacin and tamsulosin +9 mg solifenacin 6.2, 7.0 and 6.5 points, respectively), i.e., combination treatment providing improvement by 0.3–0.8 IPSS points at the group level [24]. Similarly, meta-analyses of various studies comparing the effect of a combination of an α_1_-adrenoceptor antagonist and a phosphodiesterase 5 inhibitor in men with LUTS also reported statistical superiority over either monotherapy, but the effect size estimate was only about 1 IPSS point at the group level [25,26]. Thus, benefits at the group level were less than those needed to be considered a responder at the individual patient level, regardless of which definition of a responder was used.

### 2.4. Recommendations from Regulatory Authorities

Interestingly, regulatory agencies such as US Food and Drug Administration or the European Medicine Agency provide only limited guidance on the evaluation of combination treatments. Apparently, this is motivated by the idea that testing a novel drug as add-on to standard of care should follow the same general considerations as any other trial evaluating novel treatments. Examples of these are studies in which effects of the sodium-glucose transporter 2 inhibitors were compared to placebo in patients with heart failure receiving standard of care treatment [13,14].

A different situation emerges in the development of an FDC. Regulatory authorities have issued guidelines for the non-clinical including toxicological evaluation of drug combinations [7,27] and the clinical development of FDCs [28]. The latter requires justifying the pharmacological and medical rationale for the combination for the intended therapeutic indication. Part of the rationale may be that an FDC reduces the number of administered doses and thereby improves patient adherence, but that alone is considered as insufficient rationale. Rather, it is expected that the applicant demonstrates that the FDC improves the benefit/risk ratio by increasing efficacy and/or improving safety relative to at least one of the monotherapies. However, the guideline does not inform further on the most appropriate trial designs other than stating that it can be based on placebo-controlled add-on designs, substitution of existing treatment consisting of the two concomitantly administered monotherapies with the FDC, or on initial combination treatment compared to a reference treatment, i.e., all options mentioned above. A guideline for the co-development of a combination of two or more new drugs, i.e., applicable when neither has been approved as monotherapy, states that comparing the combination to the placebo is sufficient if studies on the individual components and/or a strong rationale for the combination based on non-clinical studies are provided [29].

## 3. Treatment Benefit at the Group vs. Individual Patient Level

### 3.1. Individual Benefit/Risk Considerations

The mechanistic concept underlying combination treatment is that two or more molecular targets and their signaling cascades are involved in the pathophysiology of a given condition, and that concomitant targeting of more than one of these pathways provides greater clinical improvement than acting on either alone. For instance, male LUTSs are considered to involve a static component due to the enlargement of the prostate and a dynamic component due to contraction of prostate and urethral smooth muscle as mediated by α_1_-adrenoceptors [30]. Thus, addressing both molecular targets concomitantly can be expected to have greater effects than either monotherapy. This theory has been supported in large, double-blind, randomized, long-term (≥4 years) clinical trials for the combination of α_1_-adrenoceptor antagonist and 5α-reductase inhibitors [11,12]; of note, this combination did not provide benefit over monotherapy with the α_1_-adrenoceptor antagonist when the study duration was ≤1 year [8,9,10].

Almost all studies comparing combination treatment to monotherapy have been based on assessing data at the group level as mean or median difference between treatments. However, almost no treatment works in every patient, implying that some patients receiving combination treatment almost by definition will not experience benefits as compared to at least one of the monotherapies. For instance, studies comparing an α_1_-adrenoceptor antagonist, a 5α-reductase inhibitor and their combination reported that 39–67% of all participants were non-responders to at least one of the monotherapies [9,12]. Thus, various individual fates may occur behind an enhanced efficacy of a combination treatment at the group level, as illustrated in Figure 1. There may be some subjects that are non-responders to each monotherapy; these are unlikely to benefit from treatment with the combination of the monotherapies. It is also possible that a patient is a non-responder to one monotherapy but a responder to the other; if that person receives combination treatment, he/she will benefit from the combination but probably would benefit as much if only receiving the alternative monotherapy. For instance, IPSS improvements in men with LUTS receiving treatment with a 5α-reductase inhibitor typically are limited to those with large prostates, and even in this group, it takes at least 3–6 months to develop. Accordingly, some studies evaluating the combination of a 5α-reductase inhibitor and an α_1_-adrenoceptor antagonist and follow-up for ≤12 months found that the 5α-reductase inhibitor was only marginally better than placebo, whereas the α_1_-adrenoceptor antagonist improved IPSS considerably; combination treatments also cause a major improvement but yielded almost identical results as monotherapy with the α_1_-adrenoceptor antagonist [8,10]. Accordingly, men not sufficiently served by the α_1_-adrenoceptor antagonist had no benefit from combination treatment in this specific setting (they had benefit at the group level in studies of ≥4 years of duration [11,12]). The only group that can expect benefit from combination treatment includes patients who are (at least to some degree) responders to both monotherapies (see next section). 

On the other hand, combination treatment exposes all patients to potential harm from adverse effects of both medications, including the subjects not experiencing individual benefit. For instance, five large studies in men with LUTS comparing an α_1_-adrenoceptor antagonist (alfuzosin, doxazosin, tamsulosin, or terazosin), a 5α-reductase inhibitor (dutasteride or finasteride), and the combination of both active treatments administered for 6 to >52 months, ejaculatory abnormalities were observed more frequently in the combination than in any other group in four of these studies even more often than could be expected based on an additive effect [8,9,10,11,12]. Of note, neither of these studies found an efficacy benefit of the combination as compared to the α_1_-adrenoceptor antagonist within the first 12 months of observation, indicating that, for this comparison at least for the first year of treatment, combination did not provide benefits but caused at least some degree of harm to some patients. The resulting benefit-risk assessment may be acceptable if greater individual benefits can be achieved but less so if the benefit at the group level is at least partly due to combination treatment addressing non-responders to monotherapy. Thus, the individual benefit/risk ratio may be negative for some patients even if treatment provides benefits at the group level.

### 3.2. Expressing Outcomes as Responder Rates and Infering True Beneficiaries of Combination Treatment

Most studies evaluating combination treatments assess outcomes at the group level based on continuous variables that may or may not be normally distributed and correspondingly are expressed as means or medians [31]. However, each type of pharmacological treatment has responders and non-responders. While definitions of what constitutes a responder differ between indications or even within an indication such as male LUTS [9,12,23], responder analysis may be informative in evaluating the effects of combination treatments and in discussing the probability of a beneficial treatment outcome with patients. In the following, we will provide arguments on why expressing effects of combination treatment may be helpful. 

The percentage of responder to two monotherapies A and B and their combination can be expressed as follows:Resp T = Resp A + Resp B − Resp C,(1)
where Resp T is the hypothetical percentage of responders in the overall group of subjects exposed to combination treatment, Resp A and Resp B are the percentages of responders to monotherapy with treatments A and B, respectively, and Resp C is the percentage of subjects responding to both monotherapies (the subtraction of Resp C is required because this group by definition includes parts of Resp A and Resp B). Of note, Resp A, Resp B, and Resp T can directly be measured, whereas Resp C is a hypothetical value representing the group in which each member benefits from combination treatment because it only comprises members responsive, to some extent, to both monotherapies. If follows that all subjects not in the Resp C group do not benefit individually from combination treatments but are exposed to a risk for adverse events and, thus, potential harm. The percentage of subjects in this group P Harm can mathematically be defined as follows.
P Harm = 100 − Resp C,(2)

If Resp A, B, and T are known, Resp C can be calculated by rearrangement of Equation (1) as follows.
Resp C = Resp A + Resp B − Resp T,(3)

We can now calculate Resp T and P Harm based on various assumptions for Resp A, B, and C using Equations (1) and (2), and Resp C based on assumed or measured values of Resp A, B, and T using Equation (3). This has been performed for several examples in Table 1. Readers can make their own assumptions and/or use measured values on responder rates in monotherapies and combination treatments based on datasets of their choosing using a spreadsheet provided as Appendix A.

The clinical challenge is that Resp C is the only group that truly benefits from combination treatment because each member is responsive to both combination partners, whereas all subjects outside of Resp C and, thus, by definition part of P Harm cannot expect individual benefit but, nonetheless, are exposed to adverse event risks associated with receiving a drug that does not provide benefit to them. Although knowledge on Resp C is clinically very relevant, measuring it empirically is very challenging if not impossible. Equations (1)–(3) can be helpful to consider this issue conceptionally.

One extreme scenario is that the responder groups of the monotherapies do not overlap at all; therefore, no patient has individual benefits from the combination (see first row of Table 1). In this case, the apparent responder rate in the overall group (Resp T) would be 100% if the responder rate to either monotherapy is 50%. Nonetheless, the components of the combination treatment would have been ill chosen because no patient is likely to experience individual benefits beyond a placebo effect, but all patients receiving combination treatment are exposed to potential harm from receiving a second medication they do not need.

The other extreme scenario is that the two combination partners have almost identical responder groups (second row of Table 1), a scenario that is only realistic if the two drugs share the same molecular/cellular target. In that case, the components of the combination would also have been ill chosen (unless drug A had been underdosed) because the possible response from target A had already been fully exploited and giving more of the same will only increase a risk for adverse events. Thus, P Harm mathematically would be the 50% not being a responder to either treatment in this example, but that would be an underestimate because the responders also would be exposed to a greater risk of harm without appreciable benefits. For instance, it had been found in a large study with a maximally effective dose of tamsulosin in men with LUTS that some patients received concomitant treatment with another α_1_-adrenoceptor antagonist (not a recommended treatment option); while the study did not report efficacy data in this small subgroup (17 out of 1784 men), it found that the odds ratio to experience an adverse event relative to the overall study population was 3.872 (95% confidence interval 1.523; 9.847) [32]. Thus, optimal combination partners should be chosen to have distinct but overlapping responder groups. 

The most realistic scenario is that monotherapies at least partly address distinct responder groups that overlap to some extent. This overlapping responder group (Resp C) represents the true beneficiaries of combination treatment. Rows 3–7 of Table 1 represent various assumptions on the percentage of patients being responders to treatments A or B and to both monotherapies. These examples show that the measured responder rate in the combination arm always overestimates the percentage of subjects experiencing true benefit, the only exception being the unlikely scenario where both monotherapies have identical responder groups. In most scenarios, P Harm is larger than Resp C. How clinically relevant this is depends, of course, on the severity and incidence of adverse effects of the two monotherapies, i.e., probably less relevant for a uroselective α_1_-adrenoceptor antagonist than for a cytotoxic drug used in cancer treatment.

Resp A, B, and T can be measured in study designs where monotherapies and their combination are tested in parallel, and Resp C can be calculated from these values using Equation (3). Two of the above studies testing the combination of an α_1_-adrenoceptor antagonist and a 5α-reductase inhibitor in men with LUTS have reported responder rates for monotherapies and their combination; that they had applied slightly different definitions what a responder is [9,12], is conceptually irrelevant to the present argument. One trial compared a 6-month treatment with alfuzosin, finasteride, and their combination and found that 43%, 33%, and 42% of them, respectively, were responders (not the primary endpoint of the study) [9]. Based on comparing Resp A and Resp T, it was concluded that combination did not provide benefits. Another trial compared a 4-year treatment with tamsulosin, dutasteride, and their combination and reported about 52%, 61%, and 67% of patients to be responders, respectively [12], and concluded that combination treatment was superior to either monotherapy for IPSS improvement (neither IPSS improvement nor responder rates for such improvement were the primary endpoint of that study). Disappointingly, these data also demonstrate that 39–67% of patients were non-responders to one of the monotherapies and 33–57% were non-responders to combination treatment. Our calculations based on Equation (3) show that the true percentage of patients benefitting from dual treatment is even lower, i.e., 34% and 46%, respectively. Based on these data, we conclude that assessments at the group level are overly optimistic with regard to outcomes and thereby benefit/risk ratios at the individual patient level.

It may be argued that our above considerations are too pessimistic because we have treated responder status as yes/no characteristics whereas it is a grey scale in clinical reality. This is obviously true but misses the point. While we have used responder status as a binary variable for simplicity of the argument, it of course can also be treated as a continuous variable; this requires a more complex set of mathematical assumptions and equations but would not in principle change the conclusion. Similarly, applying the above consideration to combinations of three or more drugs (as already is standard of care in the treatment of, e.g., tuberculosis) would also require more complex equations but we feel that the same principles continue to apply. Moving on to combinations of four or more drugs may exceed the practical possibilities for testing. However, a remedy in such situations could be to consider an existing combination of two or more drugs as one treatment and the new player to be added as the second. Generally, these additional complexities would probably not change our recommendation to emphasize responder rates in the reporting of studies testing a possible superiority of combinations over simpler treatments.

It can also be argued that making responder rates the primary outcome parameter can result in a reduction in statistical power and, conversely, result in a requirement of a greater sample size that makes a study more expensive and possibly requiring more time. Therefore, we explicitly do not recommend as a routine approach to use responder rates as the primary endpoint. However, it should be calculated and reported for facilitate interpretation.

Whether exposing patients to undue risk of harm because they have limited personal benefit due to receiving a drug that does not serve them personally depends on specific aspects of seriousness of the condition to be treated on the one hand and the severity and incidence of the adverse effects of a given treatment. Therefore, benefit/risk consideration will require medical judgment. However, we propose that understanding percentages of patients truly experiencing benefit from combination treatment (Resp C) and those not receiving benefit but being exposed to an additional drug (P Harm) will aid clinical decision, which results in more than only looking at mean or median improvements at the group level.

## 4. Conclusions

We conclude that assessing benefits of combination treatment only at the group level overestimates the benefits to individual patients under almost all imaginable assumptions. Therefore, we propose that studies evaluating combination treatments should always report percentages of responders. The definition of a responder should of course be part of the study protocol and not set after data have been observed. The equations presented here may aid such efforts. Physicians should be skeptical if benefits at the group level are already small and of dubious clinical relevance because this implies that the percentage of patients truly benefitting from combination treatment is also small. Nonetheless, trying combination treatment in a given patient insufficiently response to treatment A may be an option under conditions where the existing and added drug have a benign safety profile and possible benefit/improvement of the condition can be assessed in a short period of time.

While the above considerations aid the conceptual understanding of how to assess benefit/risk ratios of combination treatments, they fail to identify specific members of the hypothetical Resp C group. Performing this will require well-validated and highly predictive biomarkers for the efficacy of either monotherapy. In a more general vein, investigators and physicians should focus on clinically meaningful effect sizes of monotherapies and combination treatments and not on the statistical significance of minor effects in large cohorts.

## Figures and Tables

**Figure 1 biomedicines-10-00270-f001:**
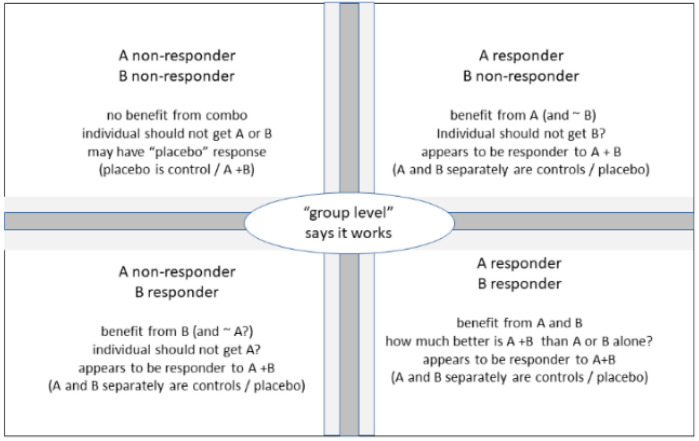
Schematic drawing of possible situations for individual patients if a drug works at the group level. Combination partners are indicated as drugs A and B. Note that this illustration is intentionally oversimplified to illustrate the concept that benefits at the population level do not necessarily imply benefits for each patient.

**Table 1 biomedicines-10-00270-t001:** Computation of Resp T and P Harm based on various assumptions for Resp A, B, and C and of Resp I based on measured values of Resp A, B, and T.

Resp A	Resp B	Resp C	Resp T	P Harm
Assumed values of A, B and C				
50	50	0	100	100
50	50	50	50	50
70	70	50	90	50
70	40	40	70	60
70	40	30	80	70
50	50	30	70	70
50	50	10	90	90
Measured values of A, B and T				
43	33	34	42	66
52	61	46	67	54

For definitions see preceding paragraph. All data are shown in % of patients exposed to combination treatment. Values in the last two rows are measured values for Resp A, B, and T as reported in [9,12]. P Harm is always a calculated value.

## Data Availability

Not applicable because this review does not report any previously undisclosed data.

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
