# Peer review of "Study Designs for Evaluation of Combination Treatment: Focus on Individual Patient Benefit"

_biomedicines, 2022, doi:10.3390/biomedicines10020270_

Round 1
Reviewer 1 Report
- Overall, complex therapy is not equally applied, and it is difficult to generalize drug treatment methods for urological diseases. Therefore, it is necessary to change the title to evaluating benefits in urological disease treatment (symptoms,), etc.
- In equation, Is it meant that Resp A and B are the percentages of responders to treatment A and B regardless of combination treatment?
Author Response
“Overall, complex therapy is not equally applied, and it is difficult to generalize drug treatment methods for urological diseases. Therefore, it is necessary to change the title to evaluating benefits in urological disease treatment (symptoms,), etc.”
It is true that combination treatments play different roles in different areas of medicine. It probably is most common in oncology and infectious diseases, but also frequent in many other areas such as obstructive airway or cardiovascular disease. The equations we have presented have no specificity for urology or any other discipline but are universally applicable. The sole reasons why our listed examples largely come from urology is that this is the therapeutic area we are most familiar with. However, our entire line of reasoning in the manuscript is independent of a therapeutic area. Therefore, we respectfully disagree with this recommendation and have decided not to adapt the manuscript title.
“In equation, is it meant that Resp A and B are the percentages of responders to treatment A and B regardless of combination treatment?”
Your interpretation is correct. Thank you for making us aware of this lack of clarity. Therefore, we have added additional wording in l. 259- 261 for clarification.
Reviewer 2 Report
The Authors classified the manuscript as "Review". They wrote"We will illustrate the discussed concepts largely
with examples from the field of functional urology (male LUTS) but propose that our considerations will be applicable to many if not most areas of medicine; a systematic review of possible examples was neither performed nor intended.(lines 38-41)" - the question is - what kind of manuscript it is. Having read this manuscript, my conclusion is - it is hard to define.
Authors discuss different study designs - parallel groups vs. add-on. Is it possible to recommend one of these design as universal? Or maybe the differences between them listed in manuscript made it impossible.
Can the equations listed in paragraph 3.2 be applied for example for the therapy with three drugs? For example the therapy of hypertension sometimes requires more than two drugs. They also should take into consideration the situation where the subjects in study respond on the treatment not only with tree drugs, but also with one or two drugs.
The proposed equations in paragraph 3.2. can be easily deduced, so I can't see the novelty.
Author Response
“The Authors classified the manuscript as "Review". They wrote "We will illustrate the discussed concepts largely with examples from the field of functional urology (male LUTS) but propose that our considerations will be applicable to many if not most areas of medicine; a systematic review of possible examples was neither performed nor intended.(lines 38-41)" - the question is - what kind of manuscript it is. Having read this manuscript, my conclusion is - it is hard to define.”
We share the feeling of the referee entirely. Originally, we had intended to submit a pure hypothesis paper, largely in line with current part 3 of the manuscript. As we did not see this category in the journal’s instructions to authors, we discussed this with the editor prior to submission. He suggested to submit under the “review” category and, where possible, adding some passages that make more review-like. The latter was implemented largely by the present part 2 of the manuscript. It probably still is a manuscript that does not easily fall into any category, but that is what the editor recommended us to do.
“Authors discuss different study designs - parallel groups vs. add-on. Is it possible to recommend one of these design as universal? Or maybe the differences between them listed in manuscript made it impossible.”
In part 2 of the manuscript, we tried to discuss the specific advantages and disadvantages of the parallel group and the add-on design. Based on these (see l. 112-121), we explicitly do not make a recommendation that one is generally superior to the other. To clarify this, we have added additional language at the end of part 2.2 (l. 122-124).
“Can the equations listed in paragraph 3.2 be applied for example for the therapy with three drugs? For example the therapy of hypertension sometimes requires more than two drugs. They also should take into consideration the situation where the subjects in study respond on the treatment not only with tree drugs, but also with one or two drugs.”
This is a great comment. We are, of course, aware that triple, quadruple or higher degrees of combination treatment already are standard of care in many areas including not only hypertension but also many malignant tumors, tuberculosis or HIV infection to name just a few. That probably would require more complex mathematical considerations, which frankly exceed our personal mathematical expertise. However, the proposed principles should probably still apply to these more complex treatment regimens. On the other hand, these added complexities do not change our recommendation to focus more on the reporting of responder rates in the evaluation of combination treatments. To address this point, we have added additional text in part 3.2 (l. 350-358).
“The proposed equations in paragraph 3.2. can be easily deduced, so I can't see the novelty.”
We fully agree that these equations can easily be deduced. When we started thinking about these, we ourselves considered them so trivial that we wondered why nobody had put them in writing before (to the best of our knowledge) or whether our reasoning was fundamentally flawed. We are happy to see that neither you nor the referee 1 felt that the equations are flawed. Perhaps, this is one of the things one considers obvious after having seen it but never thought about before. Nonetheless, the important thing is that they are correct. Of note, as far as we can see, we never claimed in the manuscript that these equations were difficult to deduce (to use an analogy: non-obviousness is a criterion to have an invention to be patentable, but obvious findings can still be very useful even in the absence of a patent).
Round 2
Reviewer 1 Report
Yes, I accepted this version .
Reviewer 2 Report
The explanations given by the Authors are satisfactory. In this case this work may be accepted.